# Earnings management and readability of CSR report: Evidence from China

**Bangqi Ren[1,2], Sheng Yao[3]***

**1** School of Economics and Management, China University of Mining and Technology, Xuzhou, China,
**2** Xuzhou Vocational Technology Academy of Finance and Economics, Xuzhou, China, **3** School of
Economics and Management, Shanghai University, Shanghai, China

\* kj9704@126.com

## Abstract

The literature has confirmed that when managers increase profits through earnings management, the readability of annual reports may be reduced Lo (2017), Ye (2018). Whether this conclusion is suitable for Chinese corporate social responsibility (CSR) reports, however, is still unclear. Based on the panel data of 5083 Chinese non-financial listed companies from 2010 to 2019, this paper adopts multiple linear regression to investigate the impact of earnings management on the readability of Chinese CSR reports. The results show that: (1) There is a significant negative correlation between earnings management and the readability of Chinese CSR reports, with the readability of Chinese annual reports as a mediating variable. (2) The negative effect is more significant when companies are not punished for violations, when the internal control index is low, when companies lack ISO14001 certification and when companies do not have independent third-party authentication for Chinese CSR reports. (3) When earnings management just exceeds zero, the readability of Chinese CSR reports decreases. (4) The economic consequences of reducing the readability of Chinese CSR reports are that financing costs are increased and environmental performance is decreased. To improve the quality of information disclosure of listed companies, the recommendations are as follows: First, the government should issue CSR reporting standards to reduce the manipulation of Chinese CSR reports. Second, Chinese CSR reports disclosed by listed companies must be audited by independent third parties to enhance the credibility of the information. Third, the company needs to strengthen its external and internal supervision to reduce the manipulation space for the readability of Chinese CSR reports. This study extends the negative relationship between earnings management and the readability from annual reports to Chinese CSR reports. To prevent investors from detecting earnings management, the readability of Chinese CSR reports may be reduced. At the same time, the study has definitely added value to the existing literature in the domain of CSR.

## Introduction

Many studies indicate that when managers increase profits through earnings management, they may reduce the readability of annual reports [1, 2]. Whether this conclusion is suitable for

**Data Availability Statement:** All data files are available from the ICPSR database (accession number(s) http://doi.org/10.3886/E197081V1).

**Funding:** This work was supported by Shanghai Philosophy and Social Science Project grant 2022BGL009. The funders had no role in study

design, data collection and analysis, decision to
publish, or preparation of the manuscript.

**Competing interests:** The authors have declared
that no competing interests exist.

Chinese CSR reports, however, is still unclear. With the development of Chinese economy, CSR reports are receiving increasing attention. More and more investors obtain business information through reading Chinese CSR reports. As a result, we want to study the relationship between earnings management and the readability of Chinese CSR reports. When managers increase profits through earnings management, how does the readability of Chinese CSR reports change?

Based on the panel data of 5083 Chinese non-financial listed companies from 2010 to 2019, our study adopts multiple linear regression to investigate the impact of earnings management on the readability of Chinese CSR reports. It is found that earnings management is negatively related to the readability of Chinese CSR reports. When managers increase profits through earnings management, they may reduce the readability of Chinese CSR reports. The negative effect is more significant when companies are not punished for violations, when the internal control index of companies is low, when companies lack ISO14001 certification and when companies do not have independent third-party authentication for Chinese CSR reports. Further study shows that when managers increase profits through earnings management, they may reduce the readability of Chinese CSR reports by decreasing the readability of Chinese annual reports. When the value of earnings management just exceeds zero, the readability of Chinese CSR reports decreases. When managers reduce the readability of Chinese CSR reports, financing costs are improved and environmental performance is decreased. To avoid endogenous effects, we use the 2SLS models and substitution variables to conduct robustness tests, and the results are still robust. The Heckman test is used to solve sample selection deviation.

Our contribution to the literature is twofold. On the one hand, Lo, Ramos, and Rogo [1] proved that the relationship between earnings management and the readability of American annual reports was negative. Ye and Wang [2] proved that the relationship between earnings management and the readability of Chinese annual reports was negative. Whether in the United States or China, managers may reduce the readability of annual reports to prevent investors from detecting artificially increased profits through earnings management. However, few studies focus on the negative relationship between earnings management and the readability of Chinese CSR reports. Therefore, we want to study the issue and extend previous research. Using theoretical and empirical methods to prove that when managers increase profits through earnings management, the readability of Chinese CSR reports may be reduced. On the other hand, we show that when managers increase profits through earnings management, that is, using accounting discretion with the aim of enhancing information asymmetry, it manifests itself as a more complex disclosure of CSR reports. Overall, we offer a more complete understanding of the relationship between earnings management and the readability of Chinese CSR reports. At the same time, this study has definitely added value to the existing literature in the domain of CSR.

The remainder of our study is organized as follows. Firstly, review the literature. Secondly, describe the research hypothesis. Thirdly, provide the research design. Fourthly, provide empirical findings. Fifthly, conclude the paper.

## Literature review

### Research on readability

There are many existing studies on corporate social responsibility and CSR performance. Pareek and Sahu [3] found a non-liner inverted U-shaped relationship between foreign ownership and the CSR performance. Ghosh, Pareek, and Sahu [4] found that board sizes may influence CSR performance. Agarwala, Pareek, and Sahu [5] found an inverse U-shape relationship

between companies' sizes and corporate social responsibility. CSR participation was positively related with small-sized firms, but as the firms became larger in size, their relationship with CSR became negative. Pareek and Sahu [6] found that there was an inverted U-shaped relationship between executive compensation and CSR performance. Ghosh, Pareek, and Sahu [7] found that governance factors like board sizes and board meetings were showing a positive effect on disclosure practices.

The readability of Chinese CSR reports is one of the studies in the field of CSR. Readability was the degree of difficulty of reading texts in a quantitative description [8]. The Flesch formula was proposed by Flesch Rudolph [9]. It used the average number of words in a sentence and the average number of syllables per word to measure the readability of a text. Gunning [8] introduced the famous Fog index, which measured the readability of a text using the average number of words and multisyllabic words. McDonald and Loughran [10] proposed measuring the readability of annual reports in terms of the size of the document. Measures of readability of English texts are not applicable to Chinese texts due to the significant differences between the two languages. Due to technical limitations, the research on Chinese readability is not as extensive as English readability. Yan and Sun [11] constructed the readability of annual reports in Chinese through manual data collections. With the rise of the Python language in China, Wang et al. [12] used transitive word density, accounting term density and subcommon word density to measure the readability of annual reports. The measurement of readability is applicable not only to annual reports but also gradually to CSR reports. Wang, Hsieh, and Sarkis [13] adopted three indices, Fog, Kincaid, and Flesch index, to measure the readability of CSR reports. Clarkson et al. [14] used total characters, total vocabularies, and keywords related to social responsibility to measure the readability of CSR reports. Jamal, Karel, and Fereshteh [15] used the length of vocabulary and the length of sentences to obtain the readability of CSR reports.

The readability of CSR reports is becoming a hot research issue, and the manipulation of the readability of CSR reports by management may lead to many economic consequences. Nazari, Hrazdil, and Mahmoudian [16] found that more readable CSR reports were associated with better CSR performance. Wang, Hsieh, and Sarkis [13] indicated that companies with stronger CSR performance were more likely to have CSR reports with greater readability. When CSR performance is better, the readability of CSR reports may be more readable. Gao et al. [17] found that the readability of CSR reports influenced investors' decision-making process. Muslu et al. [18] indicated that the better the readability of CSR reports was, the more accurate forecasts. Xu et al. [19] found that there was a positive relationship between corporate tax avoidance and the readability of CSR reports. Thuy et al. [20] found that the readability of CSR reports had a negative influence on stock price crash risk. There are many research findings on the readability of CSR reports. However, few studies foucs on the relationship between earnings management and the readability of Chinese CSR reports; therefore, we intend to study this issue.

### Research on earnings management

Schipper [21] proposed that earnings management was a purposeful intervention in external financial reporting to obtain some private interests. In capital markets, earnings management can lead to complex economic consequences. Fischer and Verrecchia [22] confirmed that company executives used earnings management to achieve their opportunistic goals, such as influencing the stock market. Botosan and Plummlee [23] considered that earnings management could reduce the cost of funding. Almahrog, Ali Aribi, and Arun [24] proved that there was a negative relationship between the level of CSR and earnings management. Ehsan et al.

[25] adopted a systematic approach to review the existing studies on the relationship between corporate social responsibility and earnings management. Managers do not want to be caught for increasing profits through earnings management. If investors detect that managers increase profits through earnings management in the capital market, the management interests may be difficult to achieve. To prevent investors from detecting such behavior, managers may reduce the readability of annual reports [1, 2]. Whether this conclusion is accurate for Chinese CSR reports, however, is still unclear. Therefore, we study the relationship between earnings management and the readability of Chinese CSR reports. In particular, when managers increase profits through earnings management, how can the readability of Chinese CSR reports be manipulated?

With the development of Chinese economy, CSR reports are increasingly valued. Chinese CSR reports not only disclose environmental information but also business information of the companies. Many investors may understand the business situation of the companies through reading Chinese CSR reports. To prevent investors from detecting earnings management, is the readability of Chinese CSR reports reduced? This issue has not been studied yet. Therefore, we are prepared to explore the issue.

## Institutional background and hypothesis development

### CSR reporting in China

In China, CSR reports are increasingly valued by stakeholder in recent years [26, 27]. More and more listed companies choose to actively disclose CSR reports. CSR reports are becoming a crucial tool to communicate with stakeholders [28]. CSR reports publish not only the social responsibility of the companies, but also the operation information [29]. Stakeholders may understand the business situation of the companies by reading Chinese CSR reports. However, the contents of Chinese CSR reports are primarily unregulated [30], thereby causing the poor CSR practice and disclosure quality of CSR reporting. Chinese CSR reports may be used to improve corporate image [31]. Additionally, if the companies fail to comply with the guidelines for disclosing information, the company will not be punished. Therefore, when managers increase profits through earnings management, the readability of Chinese CSR reports may be reduced.

### Hypothesis development

In the USA, when managers increased profits through earnings management, the readability of annual reports might be reduced [1]. In China, Ye and Wang [2] found that the readability of annual reports was worse if earnings management was greater. Whether in China or the United States, existing literature showed a negative correlation between earnings management and annual report readability. In consistency theory, individuals exhibit certain similarities and stability in their behaviors and behavioral styles in different scenarios [32, 33]. Since Chinese annual reports and Chinese CSR reports are different ways of disclosing information, managers may take consistent action to manipulate the readability of Chinese annual reports and the readability of Chinese CSR reports. The readability of Chinese CSR reports may be reduced when managers reduce the readability of Chinese annual reports according to consistency theory. Therefore, when managers increase profits through earnings management, the readability of Chinese CSR reports may be reduced.

Based on principal-agent theory, managers conduct earnings management to manipulate investors' expectation [34]. Earnings management may lead to the adverse economic consequences [35]. If investors detect that managers increase profit through earnings management, managers are regarded as violating their duties [36]. To avoid being punished by regulatory

agencies and the market, managers may take some measures to prevent their behavior from being discovered [37]. For example, investors may obtain the business information and detect earnings management through reading Chinese CSR reports. To prevent investors from detecting earnings management, managers are more motivated to disclose complex information in Chinese CSR reports. Reducing the readability of Chinese CSR reports is one of the most important means to disclose complex information. As a result, to cover up earnings management, managers may reduce the readability of Chinese CSR reports.

Based on impression management theory, the disclosure of CSR reports is not subject to the relevant standards and is voluntary in most countries. Chinese CSR reports are not constrained by specific guidelines. As a result, the readability of CSR reports is an important tool to ameliorate investors' image of the company [38]. If the artificial inflation of profits through earnings management is detected by investors, the investors' image of the company is negatively affected. To ameliorate investors' image of the company, managers may reduce the readability of Chinese CSR reports to greenwash earnings management. Based on the above discussion, our central hypothesis is as follows.

H1. When managers increase profits through earnings management, the readability of Chinese CSR reports may be reduced.

Based on legitimacy theory [39], companies could signal their legitimacy via nonfinancial disclosure after the negative effects of irregularities. When companies have irregularities, managers are not willing to reduce the readability of Chinese CSR reports. Such behavior makes it more difficult for investors to obtain information, which may further damage the image of the company's legitimacy. Therefore, when managers increase profits through earnings management, the readability of Chinese CSR reports may not be reduced. If the company has no irregularities, managers do not need to maintain the image of the company's legitimacy and may face less pressure from the SEC. Managers are more motivated to reduce the readability of Chinese CSR reports to prevent investors from detecting artificially increased profits through earnings management. Based on the above discussion, our second hypothesis is as follows.

H2. With the nonviolation penalties of companies, when managers increase profits through earnings management, the readability of Chinese CSR reports may be reduced.

When the internal control index of the companies is high, the internal control system is quite thorough. When the internal control of companies is relatively good, managers do not have much room to reduce the readability of Chinese CSR reports. In this situation, managers may disclose higher information quality in Chinese CSR reports because of a thorough internal control system [40]. The relationship between earnings management and the readability of Chinese CSR reports may not be significant. On the contrary, managers may have a greater opportunity to reduce the readability of Chinese CSR reports when the internal control index of the companies is low. Based on the above discussion, our third hypothesis is as follows.

H3. With a low internal control index of companies, when managers increase profits through earnings management, the readability of Chinese CSR reports may be reduced.

ISO14001 indicates that companies have reached the international level in environmental management, which helps them establish a positive social image [41]. To maintain a good environmental image for investors, many companies want to obtain the ISO14001 certification. However, some companies may not obtain ISO14001 certification because their environmental protection work is not very good. If companies have ISO14001 certification, the environmental protection work of the companies is well done. Managers may not reduce the readability of CSR reports to maintain a good environmental image [42]. The relationship between earnings management and the readability of Chinese CSR reports may not be significantly related. If companies do not pass the ISO14001 certification, they lack a record of complying with environmental protection. Managers are more motivated to reduce the readability

of Chinese CSR reports to prevent investors from detecting artificially increased profits through earnings management. Based on the above discussion, our fourth hypothesis is as follows.

H4. Without the ISO14001 certification of companies, when managers increase profits through earnings management, the readability of Chinese CSR reports may be reduced.

If companies have independent third-party authentication for CSR reports, Chinese CSR reports will be strictly regulated [43]. Under this situation, when managers increase profits through earnings management, the readability of Chinese CSR reports may not be decreased. On the contrary, the readability of Chinese CSR reports may be decreased if companies do not have independent third-party authentication for CSR reports. Based on the above discussion, our fifth hypothesis is as follows.

H5. Without independent third-party authentication for CSR reports, the readability of Chinese CSR reports may be decreased when managers increase profits through earnings management.

## Research design

### Data and sample selection

The data of A-share listed companies from 2010 to 2019 CSR reports is selected as the initial sample. Sample selection processes are illustrated in Table 1. Thus, we have 7,540 total observations. We exclude 95 observations that belong to ST companies. ST companies refer to a listed company that has suffered losses for two consecutive years. Next, we exclude 1,078 observations that belong to financial companies. In addition, we exclude 947 observations whose PDF documents are difficult to convert into TXT files. We cannot obtain the readability metrics from these observations. Finally, we exclude 337 observations with missing data for the control variables. We therefore have 5,083 final observations. All continuous variables are winsorized at the 1% and 99% levels. The financial data are from the China Stock Market and Accounting Research Database (CSMAR).

### Variable definitions

**The readability of CSR reports.** The readability of the Chinese CSR reports is good when a text contains simpler sentences. Simple sentences usually contain fewer characters in a single sentence and fewer vocabularies in a single sentence. The readability of the Chinese CSR reports is poor when a text contains more complex sentences. Complex sentences usually contain more characters in a single sentence and more vocabularies in a single sentence [13]. When Chinese CSR reports contain more pages, it means that more detailed information is disclosed. The readability of Chinese CSR reports is good when the texts contain more pages. Therefore, we use the average character of a single sentence, the average vocabulary of a single

**Table 1. Selection process of samples.** This table provides the process of obtaining Chinese CSR reports that need to be observed.

| Sample selection | of Obs. |
| --- | --- |
| CSR reports of all firms from 2010 to 2019 | 7540 |
| Exclude observations from ST companies | 95 |
| Exclude observations from financial companies | 1078 |
| Exclude observations that are difficult to convert from PDF to text | 947 |
| Exclude observations with control variables missing | 337 |
| Final observations | 5083 |

sentence, and the total number of pages to construct Chinese CSR readability indicators. The readability index is obtained as follows.

First, the Python programming language is used to obtain the total characters, total vocabularies, total sentences, and total pages of Chinese CSR reports. In the count of vocabularies, the "Jieba" thesaurus is used for vocabularies segmentation. Second, the average character of a single sentence and the average vocabulary of a single sentence are calculated. Third, it is necessary to homogenize the average character of a single sentence and the average vocabulary of a single sentence. The models are shown in (1).

$$
\begin{aligned}
\textit{To homogeniz the average vocabulary} \quad &(\textit{character}) = \\
&\textit{Maximum average vocabulary}(\textit{character}) \textit{ of} \\
&\textit{all samples} - \textit{Average vocabulary}(\textit{character})
\end{aligned}
\tag{1}
$$

$$
\begin{aligned}
\textit{To standardize the average vocabulary}(\textit{character}) = \\
\textit{Average vocabulary}(\textit{character}) \textit{ homogenized}/ \\
(\textit{Maximum value of all average vocabulary}(\textit{character}) \textit{ homogenized} \\
- \textit{Minimum value of all average vocabulary}(\textit{character}) \textit{ homogenized})
\end{aligned}
\tag{2}
$$

Fourth, after homogenizing the average character of a single sentence and the average vocabulary of a single sentence, we standardize the average character of a single sentence, the average vocabulary of a single sentence and the total pages. The models are shown in (2).

Fifth, we add them together to obtain the readability of Chinese CSR reports. The greater the index of readability is, the better the readability of CSR reports.

**Earnings management.** Earnings management (DA) is used to measure how much managers manipulate annual report information. A modified Jones model [35] is used to calculate DA. The models are shown in (3)–(5).

$$
\frac{TA_t}{A_{t-1}} = \alpha_1 \times \frac{1}{A_{t-1}} + \alpha_2 \times \frac{\Delta REV_t}{A_{t-1}} + \alpha_3 \times \frac{PPE_t}{A_{t-1}} + \varepsilon
\tag{3}
$$

$$
\frac{NDA_t}{A_{t-1}} = \beta_1 \times \frac{1}{A_{t-1}} + \beta_2 \times \frac{\Delta REV_t}{A_{t-1}} + \beta_3 \times \frac{PPE_t}{A_{t-1}}
\tag{4}
$$

$$
\frac{DA_t}{A_{t-1}} = \frac{TA_t}{A_{t-1}} - \frac{NDA_t}{A_{t-1}}
\tag{5}
$$

Where $TA_t$ is the total accruals, $NDA_t$ is the no discretional accruals, $DA_t$ is the manipulative accruals, $\Delta REV_t$ is the change in operating income, PPEt is the fixed asset, and $A_{t-1}$ is the total assets at the beginning of the period. Based on the cross-sectional modified Jones model, models (3) to (5) is used to estimate accrued earnings management by industry and year. When *DA* is greater, managers artificially increase more profits through earnings management.

**Control variables.** All control variables come from relevant literature on the readability of CSR reports. Following Wang, Hsieh, and Sarkis [13] and Yao [44], the control variables include common financial indicators of the firms such as *Size* (firm size), *Lev* (the ratio of liabilities to assets), *Growth* (sales growth that equals the percentage change in sales from the

previous year to the current year), *Top*1 (the shareholding ratio of the largest shareholder), *Roe* (return on equity equal to net profit over the end-of-period equity), *Cash* (monetary funds divided by current liabilities), *SOE* (state-owned enterprise display 1, nonstate owned enterprises display 0), *Dual* (if the CEO and the chairman of the board are the same person, the value is 1; otherwise, it is 0), and *Attestation* (if the Chinese CSR reports of the company are audited by independent third-party firms, the value is 1; otherwise, it is 0). In addition, we treat year and firm as dummy variables in the regressions to control for year and firm fixed effects, respectively. Besides, we present the expected sign between each variable and the readability of Chinese CSR reports based on the related literature, where "+" represents a positive correlation, "−" represents a negative correlation, and "+ /−" represents an uncertain sign. See S1 Appendix for variable definitions. All the controlling variables are obtained from the CSMAR database.

## Model design

To verify that when managers increase profits through earnings management, the readability of CSR reports may be reduced, we establish the multiple regression model (6) as follows.

$$Readability\_CSR = \alpha + \beta_1 \times DA + Controls + Firm + Year + \varepsilon \qquad (6)$$

## Empirical analysis

### Descriptive statistical analysis

Panel A of Table 2 reports the descriptive statistics for our sample. The maximum value of *Readability_CSR* is 2.122. Its minimum is 1.608, and its average is 1.864. Its standard deviation is 0.09, which indicates that the readability of Chinese CSR reports fluctuates little. Few studies have been conducted on the readability of Chinese CSR reports in domestic literature, although there are a small number of studies abroad. However, Chinese and English CSR reports readability is different because of language. As a result, studying Chinese CSR reports readability will become a supplement to existing research. The maximum value of *Readability_AR* is 0.626. Its minimum is 0.175, and its average is 0.356. The statistics of Chinese annual reports readability is basically consistent with the results of Wang et al. [12]. In the study of Wang et al. [12], the average value of *Readability_AR* is 0.784. Its standard deviation is 0.178. The maximum value of earnings management is 0.261, and its minimum is -0.213. We divide the overall sample into a high-earnings group and a low-earnings group by the mean/median of earnings management. Panel B of Table 2 reports the univariate difference test. Grouping by the mean, *Readability_CSR* in the high-DA group is 1.859. *Readability_CSR* in the low-DA group is 1.868. Grouping by the median, *Readability_CSR* in the high-DA group is 1.860, *Readability_CSR* in the low-DA group is 1.868. Panel B of Table 2 reports that the coefficient of difference between groups is at the 1% significance level. We find that the readability of CSR reports in the high-earnings group is smaller than that in the low-earnings group. Therefore, when managers artificially increase profits through earnings management, managers may reduce the readability of Chinese CSR reports. The result supports the previous assumption H1.

### Baseline regression

Table 3 reports the baseline regression results of model (6). The results show that the coefficient on earnings management and the readability of Chinese CSR reports is -0.0361, at the 1%

**Table 2. Descriptive statistics.** This table displays summary statistics. The sample period is 2010–2019, and the sample comprises 5083 firm-year observations. *Readability_CSR* is used to measure the readability of Chinese CSR reports. *Readability_CSR₁* and *Readability_CSR₂* are surrogate variables for *Readability_CSR*. *Readability_AR* is used to measure the readability of Chinese annual reports. The DA indicator variable is calculated by the modified Jones model as Models (3)–(5). Other controlling variables are obtained from the CSMAR database.

**Panel A Descriptive statistics of main variables**

| Sample | N | Min | Mean | Max | STDEV | P25 | Median | P75 |
|---|---|---|---|---|---|---|---|---|
| Readability_CSR | 5083 | 1.608 | 1.864 | 2.122 | 0.090 | 1.814 | 1.861 | 1.913 |
| Readability_CSR$_1$ | 5083 | 0.869 | 1.019 | 1.315 | 0.069 | 0.974 | 0.999 | 1.047 |
| Readability_CSR$_2$ | 5083 | 0.676 | 0.931 | 1.271 | 0.076 | 0.885 | 0.921 | 0.967 |
| Readability_AR | 5083 | 0.175 | 0.356 | 0.626 | 0.095 | 0.287 | 0.349 | 0.415 |
| DA | 5083 | -0.213 | -0.001 | 0.261 | 0.076 | -0.042 | -0.004 | 0.035 |
| Lev | 5083 | 0.065 | 0.487 | 0.860 | 0.197 | 0.342 | 0.500 | 0.638 |
| Size | 5083 | 20.520 | 23.170 | 27.050 | 1.392 | 22.130 | 23.030 | 24.040 |
| Roe | 5083 | -0.385 | 0.083 | 0.321 | 0.093 | 0.041 | 0.087 | 0.120 |
| Top1 | 5083 | 0.085 | 0.374 | 0.747 | 0.158 | 0.245 | 0.364 | 0.499 |
| Growth | 5083 | -0.635 | 0.370 | 6.283 | 0.932 | -0.028 | 0.131 | 0.413 |
| SOE | 5083 | 0.000 | 0.623 | 1.000 | 0.485 | 0.000 | 1.000 | 1.000 |
| Cash | 5083 | 0.029 | 0.600 | 6.765 | 0.933 | 0.174 | 0.317 | 0.603 |
| Dual | 5083 | 0.000 | 0.106 | 1.000 | 0.307 | 0.000 | 0.000 | 0.000 |
| Attestation | 5083 | 0.000 | 0.197 | 1.000 | 0.139 | 0.000 | 0.000 | 0.000 |

**Panel B Univariate difference test**

| Variable | Grouping | High–DA | Low–DA | Diff.Mean/Median |
|---|---|---|---|---|
| Readability_CSR | Mean | 1.859 | 1.868 | 3.5555*** |
| | Median | 1.860 | 1.868 | 3.3581*** |
| Readability_CSR$_1$ | Mean | 1.015 | 1.021 | 2.9738*** |
| | Median | 1.016 | 1.022 | 2.7484*** |
| Readability_CSR$_2$ | Mean | 0.926 | 0.934 | 3.6551*** |
| | Median | 0.927 | 0.935 | 3.4694*** |

***, **, and * indicate significance at the 1%, 5%, and 10% levels, respectively.

significance level in Column (1) for the year and firm fixed effects; -0.0375, at the 1% significance level in Column (2) for the different control variables and the firm fixed effects; -0.0545, at the 1% significance level in Column (3) for the different control variables and the year fixed effects; and -0.0374, at the 1% significance level in Column (4) for the different control variables and for the year and firm fixed effects. These results are also economically significant. In Column (1), the coefficient of -0.0361 on earnings management implies that a one-standard-deviation increase in earnings management will lead to 0.15% decrease in the average value of Chinese CSR reports readability. In Column (2), the coefficient of -0.0375 on earnings management implies that a one-standard-deviation increase in earnings management will lead to 0.15% decrease in the average value of Chinese CSR reports readability. In Column (3), the coefficient of -0.0545 on earnings management implies that a one-standard-deviation increase in earnings management will lead to 0.22% decrease in the average value of Chinese CSR reports readability. In Column (4), the coefficient of -0.0374 on earnings management implies that a one-standard-deviation increase in earnings management will lead to 0.15% decrease in the average value of Chinese CSR reports readability. The results show that when managers increase profits through earnings management, the readability of Chinese CSR reports may be reduced. Thus, we verify Hypothesis 1.

**Table 3. The impact of earnings management on the readability of Chinese CSR reports.** To test the impact of earnings management on the readability of Chinese CSR reports, we run the OLS regressions for Model (6). The regression results are listed in Columns 1 to 4 in Table 3. The dependent variable is *Readability_CSR*, measured as the readability of Chinese CSR reports. The post indicator variable is DA, measured as accrued earnings management. All the control variables are obtained from the CSMAR database, and *t*-statistics are reported in parentheses.

| Dep.Readability CSR | (1) | (2) | (3) | (4) |
|---|---|---|---|---|
| DA | -0.0361*** | -0.0375*** | -0.0545*** | -0.0374*** |
|  | (-2.90) | (-2.95) | (-3.26) | (-2.92) |
| Size |  | -0.0046* | 0.0120*** | -0.0080** |
|  |  | (-1.96) | (10.57) | (-2.32) |
| Lev |  | 0.0296** | -0.0147* | 0.0333*** |
|  |  | (2.46) | (-1.65) | (2.68) |
| Roe |  | 0.0144 | 0.0142 | 0.0197 |
|  |  | (1.21) | (1.00) | (1.61) |
| Top1 |  | 0.0499*** | -0.0190** | 0.0520*** |
|  |  | (2.75) | (-2.24) | (2.82) |
| Cash |  | 0.0021 | -0.0018 | 0.0023 |
|  |  | (1.28) | (-1.10) | (1.38) |
| SOE |  | -0.0176* | 0.0097*** | -0.0160* |
|  |  | (-1.88) | (3.49) | (-1.71) |
| Growth |  | -0.0029** | -0.0014 | -0.0031** |
|  |  | (-2.31) | (-1.05) | (-2.46) |
| Dual |  | -0.0067* | -0.0034 | -0.0077** |
|  |  | (-1.96) | (-0.85) | (-2.26) |
| Attestation |  | 0.0032 | 0.0741*** | 0.0031 |
|  |  | (0.34) | (8.31) | (0.33) |
| _cons | 1.9890*** | 2.1047*** | 1.5942*** | 2.1806*** |
|  | (107.02) | (31.35) | (67.40) | (24.17) |
| Firm | Yes | Yes | No | Yes |
| Year | Yes | No | Yes | Yes |
| N | 5083 | 5083 | 5083 | 5083 |
| Adj.R$^2$ | 0.582 | 0.584 | 0.055 | 0.585 |

***, **, and * indicate significance at the 1%, 5%, and 10% levels, respectively.

## Cross-sectional analysis

To prove different situations of earnings management on the readability of Chinese CSR reports, we divide the overall sample into four groups: whether companies have irregularities or not [39], the value of the internal control index [40], whether companies have ISO14001 or not [41, 42] and whether companies have independent third-party authentication for CSR reports or not to conduct cross-sectional analysis [43]. Table 4 reports the impact of earnings management on the readability of Chinese CSR reports under different situations.

Panel A of Table 4 reports the regression results of model (6) based on whether companies have irregularities [39]. When companies have irregularities, the value is 1; otherwise, it is 0. The result shows that when companies do not have irregularities, the negative effect between earnings management and the readability of Chinese CSR reports is significant (-0.0402, at the 1% significance level). However, for companies with irregularities, the negative effect between earnings management and the readability of Chinese CSR reports is not significant. The group different coefficient tests are significant (-0.0402, at the 1% significance level). The regression

**Table 4. Cross-sectional analysis.** This table of Panel A displays the results of regression estimations of the relationship between earnings management and the readability of Chinese CSR reports, dividing the sample into irregular and nonirregular firms. The irregularities indicator variable equals 1 if companies have irregularities; otherwise, it is 0. Panel B displays the results of regression estimations of the relationship between earnings management and the readability of Chinese CSR reports, dividing the sample into a high internal control index and a low internal control index. We choose the internal control index of the DIB database. When the value is greater than the median of all the samples, the value is 1; otherwise, it is 0. Panel C displays the results of regression estimations of the relationship between earnings management and the readability of Chinese CSR reports, dividing the sample into ISO14001 and non-ISO14001 firms. The value is 1 when the company has passed the ISO14001 certification; otherwise, it is 0. Panel D displays the results of regression estimations of the relationship between earnings management and the readability of Chinese CSR reports, dividing the sample into attestation and non- attestation. If companies have independent third-party authentication for CSR reports, the value is 1; otherwise, it is 0. The dependent variable is *Readability_CSR* measured as the readability of Chinese CSR reports. The post indicator variable is DA, measured as accrued earnings management. All the controlling variables are obtained from the CSMAR database, and t-statistics are reported in parentheses.

**Panel A: Irregularities**

| Dep.Readability_CSR | Irregularities | Nonirregularities |
|---|---|---|
| DA | -0.0141 | -0.0402*** |
| | (-0.21) | (-2.95) |
| Controls | Yes | Yes |
| Firm | Yes | Yes |
| Year | Yes | Yes |
| Adj.$R^2$ | 0.786 | 0.575 |
| N | 390 | 4693 |
| Group difference coefficient test | -0.0402*** | |

**Panel B: Internal control index**

| Dep.Readability_CSR | High | Low |
|---|---|---|
| DA | -0.0274 | -0.0403** |
| | (-1.15) | (-2.47) |
| Controls | Yes | Yes |
| Firm | Yes | Yes |
| Year | Yes | Yes |
| Adj.$R^2$ | 0.633 | 0.580 |
| N | 2102 | 2981 |
| Group difference coefficient test | -0.0403** | |

**Panel C: ISO14001**

| Dep Readability_CSR | ISO14001 | Non–ISO14001 |
|---|---|---|
| DA | 0.0337 | -0.0420*** |
| | (0.77) | (-3.08) |
| Controls | Yes | Yes |
| Firm | Yes | Yes |
| Year | Yes | Yes |
| Adj.$R^2$ | 0.615 | 0.594 |
| N | 568 | 4515 |
| Group difference coefficient test | -0.0419*** | |

**Panel D: Attestation**

| Dep.Readability_CSR | Attestation | Non attestation |
|---|---|---|
| DA | 0.2708 | -0.0383*** |
| | (0.79) | (-3.02) |
| Controls | Yes | Yes |
| Firm | Yes | Yes |
| Year | Yes | Yes |
| Adj.$R^2$ | 0.342 | 0.583 |
| N | 100 | 4983 |
| Group difference coefficient test | -0.0383*** | |

***, **, and * indicate significance at the 1%, 5%, and 10% levels, respectively.

results indicate that if companies with irregularities, managers may not reduce the readability of Chinese CSR reports due to maintaining the image of companies' legitimacy. When companies do not have irregularities, managers may not maintain the image of companies' legitimacy. When managers increase profits through earnings management, the readability of Chinese CSR reports may be reduced. Therefore, we verify Hypothesis 2.

Panel B of Table 4 reports the regression results of model (6) based on the value of the internal control index [40]. We choose the internal control index from the DIB database. When the value is greater than the median of all the samples, the value is 1; otherwise, it is 0. The result shows that when companies belong to the low internal control index group, the negative effect between earnings management and the readability of Chinese CSR reports is significant (-0.0403, at the 5% significance level). However, when companies belong to the high internal control index group, the negative effect between earnings management and the readability of Chinese CSR reports is not significant. The group different coefficient tests are significant (-0.0403, at the 5% significance level). The regression result indicates that when companies have a perfect internal control system, managers may not have room to reduce the readability of Chinese CSR reports. When the internal control of companies is defective, managers have opportunities to reduce the readability of Chinese CSR reports to prevent investors from detecting artificially increased profits through earnings management. Therefore, we verify Hypothesis 3.

Panel C of Table 4 reports the regression results of model (6) based on whether companies have ISO14001 or not [41, 42]. The value is 1 when the company has passed the ISO14001 certification; otherwise, it is 0. The results show that when companies do not have ISO14001 certification, the negative effect between earnings management and the readability of Chinese CSR reports is significant (-0.0420, at the 1% significance level). However, when companies have ISO14001 certification, the negative effect between earnings management and the readability of Chinese CSR reports is not significant. The group different coefficient tests are significant (-0.0419, at the 1% significance level). The regression results indicate that if companies have ISO14001 certification, they have a good impact on environmental protection. Managers are willing to disclose accurate information without reducing the readability of Chinese CSR reports. However, if companies do not have ISO14001 certification, it means that environmental protection of companies is not done well. They may be more motivated to reduce the readability of Chinese CSR reports to prevent investors from detecting artificially increased profits through earnings management. Therefore, we verify Hypothesis 4.

Panel D of Table 4 reports the regression results of model (6) based on whether companies have independent third-party authentication for Chinese CSR reports or not [43]. We use Attestation to measure whether a listed company is independent audited for CSR reports. The value is 1 when the company has independent third-party authentication for Chinese CSR reports; otherwise, it is 0. The results show that when companies do not have independent third-party authentication for Chinese CSR reports, the negative effect between earnings management and the readability of Chinese CSR reports is significant (-0.0383, at the 1% significance level). However, when companies have independent third-party authentication for Chinese CSR reports, the negative effect between earnings management and the readability of Chinese CSR reports is not significant. The regression results indicate that if companies do not have independent third-party authentication for Chinese CSR reports, Chinese CSR reports are not strictly regulated. When managers increase profits through earnings management, the readability of Chinese CSR reports may be decreased. On the contrary, the readability of Chinese CSR reports may not be decreased if companies have independent third-party authentication for CSR reports. Because Chinese CSR reports are strictly regulated, managers may not reduce the readability of Chinese CSR reports. Therefore, we verify Hypothesis 5.

## Robustness test

**Consideration of endogenous problems.** The explained variable of this paper is the readability of Chinese CSR reports. The core explanatory variable is earnings management. It has been proven that earnings management has a negative relationship with the readability of Chinese CSR reports, but earnings management and the readability Chinese CSR reports may interact, resulting in endogeneity problems [24, 25]. Therefore, we use the 2SLS to solve the endogeneity problems in this paper. We choose Aggressiveness as the tool variable in this paper. Hamilton et al. [45] have proven that Aggressiveness is positive with earnings management. Therefore, Aggressiveness is related to earnings management. Firstly, the company's net profit for the year minus the company's net cash flow from operating activities for the year, then we obtain the result. We use the above result to divide by the company's total assets at the beginning of the year. Finally, Aggressiveness is obtained. In terms of the calculation process, all data are from annual reports, and there is no relevance to CSR reports. As a result, Aggressiveness is not related to the readability of CSR reports. We choose Aggressiveness as a tool variable to perform the 2SLS.

Column (1) of Table 5 indicates that the coefficient on Aggressiveness and earnings management is 0.1248 at the 1% significance level. Column (2) of Table 5 shows that the coefficient on earnings management and the readability of CSR reports is -0.1036 at the 10% significance level. The results are consistent with the main regression results, again verifying Hypothesis 1.

**Sample self-selection test.** Chinese CSR reports are not disclosed by all companies. Therefore, the disclosure of Chinese CSR reports does not happen randomly. There is a problem of sample self-selection. To solve this problem, the Heckman two-stage model is used [46]. The value of Disclosure is 1 if the Chinese CSR report of the company is disclosed; otherwise, it is 0. All listed companies from 2010 to 2019 is selected as the whole sample. To eliminate ST companies and missing control variables, we finally obtain 21,498 observations. In China, if the companies belong to the heavy-pollution industry, the disclosure of CSR reports is mandatory. If the companies belong to the nonheavy-pollution industry, the disclosure of CSR reports is voluntary. Whether a company belongs to a heavy-pollution industry may influence the disclosure of CSR reports. However, depending on whether a company belongs to a heavy-pollution industry or a nonheavy-pollution industry, managers may increase or reduce the readability of Chinese CSR reports. Pollution may influence the disclosure of CSR reports but may not influence the readability of CSR reports [47]. Therefore, we choose Pollution as a tool variable.

The Heckman two-stage model is used [48] to solve the problems of sample self-selection. The first stage is the probit model to make use of the total sample to calculate the probability of Chinese CSR report disclosure. And the IMR of each observation value is calculated. The second stage is to use the OLS model to study the relationship between earnings management and the readability of Chinese CSR reports. The IMR of the first stage is regarded as the control variable.

Column (2) of Table 6 shows that the coefficients on IMR are -0.0180 at the 10% significance level. There is a problem of sample self-selection. Column (1) of Table 6 shows that the coefficients on Pollution and Disclosure are 2.3365 at the 1% significance level. Column (2) of Table 6 shows that the coefficient on earnings management and the readability of Chinese CSR reports is -0.0338 at the 1% significance level. The results indicate that sample self-selection has little impact on the conclusion. The results are consistent with the main regression results, and we again verify Hypothesis 1.

**Variable substitution.** *Replace dependent variable.* By adding the average character of a single sentence and total pages of Chinese CSR reports, the readability of CSR reports named

**Table 5. Consideration of endogenous problems.** Earnings aggressiveness is a tool variable for performing 2SLS to solve endogeneity problems. The dependent variable is *Readability_CSR*, measured as the readability of Chinese CSR reports. The post indicator variable is DA, measured as accrued earnings management. All the controlling variables are obtained from the CSMAR database, and *t*-statistics are reported in parentheses.

| Dep. | The first stage DA | The second stage Readability |
|---|---|---|
| **DA** | | -0.1036* |
| | | (-1.87) |
| **Aggressiveness** | 0.1248*** | |
| | (14.36) | |
| **Size** | 0.0087** | -0.0075** |
| | (2.18) | (-2.35) |
| **Lev** | -0.0682*** | 0.0294** |
| | (-4.72) | (2.47) |
| **Roe** | 0.2028*** | 0.0342** |
| | (14.58) | (2.09) |
| **Top1** | 0.0027 | 0.0519*** |
| | (0.12) | (3.05) |
| **Cash** | 0.0006 | 0.0024 |
| | (0.32) | (1.60) |
| **SOE** | -0.0114 | -0.0167* |
| | (-1.05) | (-1.92) |
| **Growth** | 0.0015 | -0.0030** |
| | (1.01) | (-2.57) |
| **Dual** | 0.0053 | -0.0073** |
| | (1.33) | (-2.32) |
| **Attestation** | -0.0021 | 0.0033 |
| | (-0.20) | (0.38) |
| **_cons** | -0.2732*** | 2.1646*** |
| | (-2.61) | (25.68) |
| **Firm** | Yes | Yes |
| **Year** | Yes | Yes |
| **N** | 5083 | 5083 |
| **Adj.R$^2$** | 0.222 | 0.582 |

***, **, and * indicate significance at the 1%, 5%, and 10% levels, respectively.

*Readability_CSR*$_1$ is remeasured. Table 7 reports the regression results. The results show that the coefficient on earnings management and the readability of Chinese CSR reports is -0.0200, at the 1% significance level in Column (1) for the year and firm fixed effects; -0.0227, at the 1% significance level in Column (2) for the different control variables and the firm fixed effects; -0.0402, at the 1% significance level in Column (3) for the different control variables and the year fixed effects; and -0.0237, at the 1% significance level in Column (4) for the different control variables and for the year and firm fixed effects.

We add the average vocabulary of a single sentence and total pages of CSR reports to remeasure the readability of Chinese CSR reports named *Readability_CSR*$_2$. Table 8 reports the regression results. The results show that the coefficient on earnings management and the readability of Chinese CSR reports is -0.0292, at the 1% significance level in Column (1) for the year and firm fixed effects; -0.0306, at the 1% significance level in Column (2) for the different control variables and the firm fixed effects; -0.0486, at the 1% significance level in Column (3)

**Table 6. Sample self-selection test.** We use Pollution as a tool variable to perform Heckman regression. The dependent variable is *Readability_CSR*, measured as the readability of Chinese CSR reports. The post indicator variable is DA, measured as accrued earnings management. All the controlling variables are obtained from the CSMAR database, and t-statistics are reported in parentheses.

| Dep. | The first stage Disclosure | The second stage Readability |
|---|---|---|
| **DA** | | -0.0338*** |
| | | (-2.76) |
| **Pollution** | 2.3365*** | |
| | (26.33) | |
| **Size** | 0.2121*** | -0.0101*** |
| | (24.18) | (-2.79) |
| **Lev** | -0.2267*** | 0.0339*** |
| | (-4.13) | (2.79) |
| **Roe** | 0.1435** | 0.0118 |
| | (1.98) | (1.23) |
| **Top1** | 0.0525 | 0.0521*** |
| | (0.81) | (2.83) |
| **Cash** | -0.0449*** | 0.0017 |
| | (-7.29) | (1.51) |
| **SOE** | 0.1862*** | -0.0190** |
| | (9.17) | (-1.99) |
| **Growth** | -0.0460*** | -0.0021** |
| | (-6.97) | (-2.00) |
| **Dual** | 0.0208 | -0.0079** |
| | (0.66) | (-2.32) |
| **Attestation** | 0.3712** | 0.0012 |
| | (2.52) | (0.13) |
| **IMR** | | -0.0180* |
| | | (-1.73) |
| **_cons** | -4.8339*** | 2.2471*** |
| | (-26.27) | (23.14) |
| **Firm** | No | Yes |
| **Year** | No | Yes |
| **N** | 21498 | 5083 |
| **Adj.R$^2$** | 0.1419 | 0.585 |

***, **, and * indicate significance at the 1%, 5%, and 10% levels, respectively.

for the different control variables and the year fixed effects; and -0.0311, at the 1% significance level in Column (4) for the different control variables and for the year and firm fixed effects. The results of Tables 7 and 8 indicate that when managers artificially increase profits through earnings management, the readability of Chinese CSR reports may be reduced. Our central hypothesis has been confirmed again.

*Replace the independent variable.* We use real earnings management to replace accrued earnings management for regression analysis [49]. Table 9 reports the regression results. The results show that the coefficient on earnings management and the readability of Chinese CSR reports is -0.0162, at the 1% significance level in Column (1) for the year and firm fixed effects; -0.0179, at the 1% significance level in Column (2) for the different control variables and the firm fixed effects; -0.0240, at the 1% significance level in Column (3) for the different control

**Table 7. Substitute dependent variables.** We use *Readability_CSR₁* to replace *Readability_CSR* for OLS regression. The regression results are listed in Table 7. The dependent variable is the readability of Chinese CSR reports. The post indicator variable is DA, measured as accrued earnings management. All the controlling variables are obtained from the CSMAR database, and t-statistics are reported in parentheses.

| Dep.Readability_CSR₁ | (1) | (2) | (3) | (4) |
|---|---|---|---|---|
| DA | -0.0200*** | -0.0227*** | -0.0402*** | -0.0237*** |
| | (-2.82) | (-3.12) | (-3.50) | (-3.25) |
| Size | | 0.0156*** | 0.0190*** | 0.0047** |
| | | (11.48) | (24.25) | (2.39) |
| Lev | | -0.0022 | -0.0294*** | 0.0104 |
| | | (-0.32) | (-4.82) | (1.47) |
| Roe | | 0.0062 | 0.0243** | 0.0164** |
| | | (0.92) | (2.47) | (2.35) |
| Top1 | | -0.0098 | -0.0020 | 0.0007 |
| | | (-0.94) | (-0.34) | (0.07) |
| Cash | | 0.0009 | -0.0017 | 0.0018* |
| | | (1.02) | (-1.51) | (1.92) |
| SOE | | -0.0182*** | 0.0037* | -0.0151*** |
| | | (-3.40) | (1.93) | (-2.83) |
| Growth | | -0.0012 | -0.0030*** | -0.0013* |
| | | (-1.60) | (-3.25) | (-1.80) |
| Dual | | -0.0050** | -0.0055** | -0.0056*** |
| | | (-2.57) | (-1.97) | (-2.89) |
| Attestation | | 0.0194*** | 0.1076*** | 0.0201*** |
| | | (3.68) | (17.50) | (3.82) |
| _cons | 1.1244*** | 0.7476*** | 0.5760*** | 1.0079*** |
| | (106.19) | (19.45) | (35.34) | (19.65) |
| Firm | Yes | Yes | No | Yes |
| Year | Yes | No | Yes | Yes |
| N | 5083 | 5083 | 5083 | 5083 |
| Adj.R² | 0.763 | 0.761 | 0.215 | 0.765 |

***, **, and * indicate significance at the 1%, 5%, and 10% levels, respectively.

variables and the year fixed effects; and -0.0178, at the 1% significance level in Column (4) for the different control variables and for the year and firm fixed effects. The results indicate that when managers artificially increase profits through earnings management, the readability of Chinese CSR reports may be reduced. The central hypothesis has been confirmed again.

## Further studies

**Impact path analysis.** Based on principal-agent theory and impression management theory, when excessive earnings management is detected by investors, they may conclude that the company is not well run and develop a bad impression of the company [34, 38]. To prevent investors from detecting artificially increased profits through earnings management, Lo, Ramos, and Rogo [1] proved that when managers increased profits through earnings management, the readability of American annual reports may be reduced. Ye and Wang [2] proved that when managers increased profits through earnings management, the readability of Chinese annual reports may be reduced. If investors cannot detect artificially inflated profits through earnings management from Chinese annual reports, they will pay more attention to

**Table 8. Substitute dependent variables.** We use *Readability_CSR$_2$* to replace *Readability_CSR* for OLS regression. The regression results are listed in Table 8. The dependent variable is the readability of Chinese CSR reports. The post indicator variable is DA, measured as accrued earnings management. All the controlling variables are obtained from the CSMAR database, and t-statistics are reported in parentheses.

| Dep.Readability_CSR$_2$ | (1) | (2) | (3) | (4) |
|---|---|---|---|---|
| DA | -0.0292*** | -0.0306*** | -0.0486*** | -0.0311*** |
| | (-3.10) | (-3.17) | (-3.61) | (-3.20) |
| Size | | 0.0031* | 0.0151*** | -0.0023 |
| | | (1.75) | (16.42) | (-0.89) |
| Lev | | 0.0167* | -0.0204*** | 0.0228** |
| | | (1.83) | (-2.85) | (2.42) |
| Roe | | 0.0094 | 0.0175 | 0.0156* |
| | | (1.04) | (1.51) | (1.68) |
| Top1 | | 0.0259* | -0.0156** | 0.0305** |
| | | (1.89) | (-2.28) | (2.18) |
| Cash | | 0.0017 | -0.0014 | 0.0021* |
| | | (1.40) | (-1.12) | (1.69) |
| SOE | | -0.0184*** | 0.0065*** | -0.0165** |
| | | (-2.59) | (2.86) | (-2.33) |
| Growth | | -0.0021** | -0.0019* | -0.0022** |
| | | (-2.17) | (-1.77) | (-2.32) |
| Dual | | -0.0057** | -0.0039 | -0.0065** |
| | | (-2.22) | (-1.21) | (-2.51) |
| Attestation | | 0.0116* | 0.0788*** | 0.0118* |
| | | (1.66) | (10.93) | (1.68) |
| _cons | 1.0493*** | 0.9763*** | 0.5881*** | 1.1039*** |
| | (74.51) | (19.18) | (30.78) | (16.15) |
| Firm | Yes | Yes | No | Yes |
| Year | Yes | No | Yes | Yes |
| N | 5083 | 5083 | 5083 | 5083 |
| Adj.R$^2$ | 0.651 | 0.652 | 0.104 | 0.653 |

***, **, and * indicate significance at the 1%, 5%, and 10% levels, respectively.

Chinese CSR reports. As an important channel for investors to obtain information, CSR reports include not only environmental information but also business information [29]. When managers inflate profits through earnings management, the readability of Chinese CSR reports may be reduced. Whether this leads to reducing the readability of Chinese CSR reports through decreasing the readability of Chinese annual reports, however, is still unclear.

Lo, Ramos, and Rogo [1] have proved that managers increase profits through earnings management, the readability of American annual reports may be reduced. Ye and Wang [2] have proved that when managers increase profits through earnings management, the readability of Chinese annual reports may be reduced. Chinese CSR reports not only disclose the companies' social responsibility, but also the companies' operation [29]. Investors may obtain the information through reading Chinese CSR reports. If managers increase profits through earnings management, investors may detect such behavior through reading Chinese CSR reports. To prevent investors from detecting such situation, the readability of Chinese CSR reports may be reduced by managers. Based on consistency theory, reducing the readability of Chinese annual reports may lead to the decrease of the readability of Chinese CSR reports. As a result, the readability of Chinese annual reports may become intermediary variable between earnings

**Table 9. Substitute independent variables.** We use Rem to replace DA for OLS regression. The regression results are listed in Table 9. The dependent variable is the readability of Chinese CSR reports. The post indicator variable is Rem, measured as real earnings management. All the controlling variables are obtained from the CSMAR database, and *t*-statistics are reported in parentheses.

| Dep.Readability_CSR | (1) | (2) | (3) | (4) |
|---|---|---|---|---|
| **Rem** | -0.0162*** | -0.0179*** | -0.0240*** | -0.0178*** |
| | (-2.88) | (-3.11) | (-3.52) | (-3.08) |
| **Size** | | -0.0044* | 0.0119*** | -0.0076** |
| | | (-1.86) | (10.50) | (-2.22) |
| **Lev** | | 0.0328*** | -0.0081 | 0.0363*** |
| | | (2.72) | (-0.91) | (2.92) |
| **Roe** | | -0.0008 | -0.0100 | 0.0042 |
| | | (-0.07) | (-0.69) | (0.34) |
| **Top1** | | 0.0516*** | -0.0187** | 0.0536*** |
| | | (2.85) | (-2.20) | (2.90) |
| **Cash** | | 0.0021 | -0.0016 | 0.0022 |
| | | (1.28) | (-1.03) | (1.37) |
| **SOE** | | -0.0179* | 0.0098*** | -0.0163* |
| | | (-1.92) | (3.52) | (-1.74) |
| **Growth** | | -0.0032** | -0.0017 | -0.0035*** |
| | | (-2.57) | (-1.26) | (-2.74) |
| **Dual** | | -0.0068** | -0.0040 | -0.0079** |
| | | (-2.01) | (-1.00) | (-2.30) |
| **Attestation** | | 0.0032 | 0.0746*** | 0.0030 |
| | | (0.34) | (8.36) | (0.33) |
| **_cons** | 1.9906*** | 2.1012*** | 1.5961*** | 2.1753*** |
| | (107.18) | (31.28) | (67.59) | (24.09) |
| **Firm** | Yes | Yes | No | Yes |
| **Year** | Yes | No | Yes | Yes |
| **N** | 5083 | 5083 | 5083 | 5083 |
| **Adj.R²** | 0.582 | 0.584 | 0.055 | 0.585 |

***, **, and * indicate significance at the 1%, 5%, and 10% levels, respectively.

management and the readability of Chinese CSR reports. We use the mediation effect test process to test the readability of Chinese annual reports [50]. All annual reports are downloaded to obtain the total characters of every Chinese annual report using Python. Wang et al. [12] indicated that the readability of Chinese annual reports was poor because of the greater number of characters. Therefore, we use total characters of Chinese annual reports after standardizing and homogenizing to structure the readability of Chinese annual reports. By using character of every Chinese annual report divided by average character of all the Chinese annual reports, the standardized characters are obtained. To obtain Chinese annual reports readability, we take the reciprocal of standardizing characters. It is called the homogenization [51]. Panel A of Table 10 shows that the coefficient on earnings management and the readability of Chinese CSR reports is -0.0374, at the 1% significance level in Column (1). The coefficient on earnings management and the readability of Chinese annual reports is -0.0573, at the 5% significance level in Column (2). The coefficient on the readability of Chinese annual reports and the readability of Chinese CSR reports is not significant in Column (3). From Columns (1) to (3) of Panel B of Table 10, we use bootstrap tests and 500 times, 800 times, and

**Table 10. The mediating effect of the readability of annual reports.** This table of Panel A displays the results of the mediating effect between earnings management and the readability of Chinese CSR reports. The dependent variable is *Readability_CSR*, measured as the readability of Chinese CSR reports. The post indicator variable is DA, measured as accrued earnings management. The mediating variable is *Readability_AR*, measured as the readability of Chinese annual reports.

| Panel A The mediating effect of the readability of annual reports | | | |
|---|---|---|---|
| Dep | Readability_CSR | Readability_AR | Readability_CSR |
| DA | -0.0374*** | -0.0573** | -0.0371*** |
| | (-2.92) | (-2.23) | (-2.89) |
| Readability_AR | | | 0.0063 |
| | | | (0.83) |
| Size | -0.0080** | -0.1343*** | -0.0071** |
| | (-2.32) | (-19.53) | (-1.99) |
| Lev | 0.0333*** | -0.0088 | 0.0333*** |
| | (2.68) | (-0.35) | (2.68) |
| Roe | 0.0197 | -0.0118 | 0.0198 |
| | (1.61) | (-0.48) | (1.61) |
| Top1 | 0.0520*** | 0.0266 | 0.0518*** |
| | (2.82) | (0.72) | (2.81) |
| Cash | 0.0023 | 0.0035 | 0.0022 |
| | (1.38) | (1.05) | (1.37) |
| SOE | -0.0160* | 0.0115 | -0.0161* |
| | (-1.71) | (0.61) | (-1.72) |
| Growth | -0.0031** | -0.0010 | -0.0031** |
| | (-2.46) | (-0.40) | (-2.46) |
| Dual | -0.0077** | 0.0097 | -0.0078** |
| | (-2.26) | (1.42) | (-2.27) |
| Attestation | 0.0031 | -0.0601*** | 0.0035 |
| | (0.33) | (-3.24) | (0.37) |
| _cons | 2.1806*** | 4.1756*** | 2.1542*** |
| | (24.17) | (23.08) | (22.52) |
| Firm | Yes | Yes | Yes |
| Year | Yes | Yes | Yes |
| N | 5083 | 5083 | 5083 |
| Adj.R$^2$ | 0.585 | 0.847 | 0.585 |
| Panel B Bootstrap method test | | | |
| | 500times | 800times | 1000times |
| BC | (0.0004,0.0041) | (0.0004,0.0044) | (0.0005,0.0048) |
| P | (0.0004,0.0040) | (0.0003,0.0042) | (0.0003,0.0042) |

***, **, and * indicate significance at the 1%, 5%, and 10% levels, respectively.

1,000 times repeated sampling excluding 0. The symbol of -0.0573*0.0063 is the same as -0.0371. As a result, a partial intermediary effect of *Readability_AR* is established.

**Economic consequences.** When managers inflate profits through earnings management, they may reduce the readability of Chinese CSR reports, which increases information asymmetry. Complex information disclosure makes investors afraid to invest [52] and makes creditors cautious about borrowing money [53]. Following Tran [54], we construct the indicator of debt financing cost. Following Easton [55], we construct the indicator of equity financing cost. Therefore, when managers reduce the readability of Chinese CSR reports, debt financing costs and equity financing costs may increase. When the readability of CSR reports is greater, CSR

performance will be better [15]. Therefore, the relationship between the readability of Chinese CSR reports and CSR performance may be positive. When managers reduce the readability of Chinese CSR reports, CSR performance may be reduced.

Table 11 shows that the coefficient on the readability of Chinese CSR reports and debt financing costs is -0.0113, at the 5% significance level in Column (1). This means that when managers reduce the readability of Chinese CSR reports, debt financing costs are increased because of information asymmetry. Table 11 shows that the coefficient on the readability of Chinese CSR reports and equity financing costs is -0.1074, at the 1% significance level in Column (2). This indicates that when managers reduce the readability of Chinese CSR reports, equity financing costs increase because of information asymmetry. When managers reduce the readability of Chinese CSR reports, investors and creditors are hard to obtain objective information through reading Chinese CSR reports. Because of information asymmetry, investors

**Table 11. Economic consequences of the readability of CSR reports.** The dependent variables are debt costs and equity costs, measured as the financing costs of the company. The dependent variable is CSR performance, measured as social welfare expenditure divided by total number of shares. The post indicator variable is *Readability*, measured as the readability of Chinese CSR reports. All the controlling variables are obtained from the CSMAR database, and *t*-statistics are reported in parentheses.

| Dep. | Debtcosts | Equitycosts | CSR performance |
|---|---|---|---|
| Readability | -0.0113** | -0.1074*** | 0.8430** |
|  | (-2.12) | (-2.80) | (2.49) |
| Average Character | -0.0130 | 0.1622 | -1.3911 |
|  | (-0.25) | (0.43) | (-0.42) |
| Average Vocabulary | 0.0177 | 0.0452 | -1.0511 |
|  | (0.78) | (0.28) | (-0.73) |
| Size | -0.0008 | 0.0259*** | 0.0898** |
|  | (-1.20) | (5.37) | (2.10) |
| Lev | 0.0358*** | -0.0191 | 0.2300 |
|  | (14.85) | (-1.10) | (1.50) |
| Roe | -0.0107*** | 0.1398*** | 0.7075*** |
|  | (-4.62) | (8.36) | (4.79) |
| Top1 | -0.0026 | -0.0133 | -0.0161 |
|  | (-0.74) | (-0.51) | (-0.07) |
| Cash | -0.0061*** | -0.0029 | 0.0016 |
|  | (-19.15) | (-1.29) | (0.08) |
| SOE | -0.0048*** | 0.0003 | -0.0958 |
|  | (-2.61) | (0.02) | (-0.83) |
| Growth | -0.0001 | -0.0006 | -0.0053 |
|  | (-0.46) | (-0.33) | (-0.34) |
| Dual | -0.0006 | 0.0094** | -0.0496 |
|  | (-0.91) | (1.97) | (-1.17) |
| Attestation | -0.0052*** | -0.0066 | 0.1701 |
|  | (-2.89) | (-0.51) | (1.49) |
| _cons | 0.0229 | -0.5197** | -2.1090 |
|  | (0.64) | (-2.01) | (-0.92) |
| Firm | Yes | Yes | Yes |
| Year | Yes | Yes | Yes |
| N | 5083 | 5083 | 5083 |
| Adj.R$^2$ | 0.676 | 0.262 | 0.638 |

***, **, and * indicate significance at the 1%, 5%, and 10% levels, respectively

may be cautious about investing and creditors may be afraid of borrowing money. As a result, when managers reduce the readability of Chinese CSR reports, financing costs are increased. Table 11 shows that the coefficient on the readability of Chinese CSR reports and CSR performance is 0.8430, at the 5% significance level in Column (3). The result indicates that when managers decrease the readability of Chinese CSR reports, CSR performance may be reduced.

**The readability of Chinese CSR reports jumps down.**   When the value of earnings management is zero, managers may not manipulate profits artificially. When earnings management just exceeds zero, managers are more motivated to increase profits artificially to influence investors' decisions. Managers do not want investors to detect the artificially increased profits through reading Chinese CSR reports. When earnings management just exceeds zero, managers may reduce the readability of Chinese CSR reports to prevent investors from detecting such behaviors. As a result, we choose the zero point of earnings management as the breakpoint [56] and perform regression discontinuity.

Fig 1 indicates that when earnings management just exceeds zero, the readability of Chinese CSR reports jumps downward. When earnings management just exceeds zero, managers may reduce the readability of Chinese CSR reports. Table 12 shows that the coefficient on DA and the readability of Chinese CSR reports is -2.38e-16 at the 1% significance level. When managers increase profits through earnings management, they may reduce the readability of Chinese CSR reports. Our central hypothesis has been confirmed again.

## Conclusions

### Unique contributions

The study of the readability of Chinese CSR reports has definitely added value to the existing literature in the domain of CSR. Existing literature proved that managers decreased the readability of annual reports to prevent investors from detecting artificially increased profits through earnings management [1, 2]. We extend the conclusion of the study from annual reports to Chinese CSR reports. When managers increase profits through earnings management, they may reduce the readability of Chinese CSR reports. The negative effect is more significant when companies are not punished for violations, when the internal control index of companies is low, when companies lack ISO14001 certification and when companies do not have independent third-party authentication for Chinese CSR reports. Further study shows

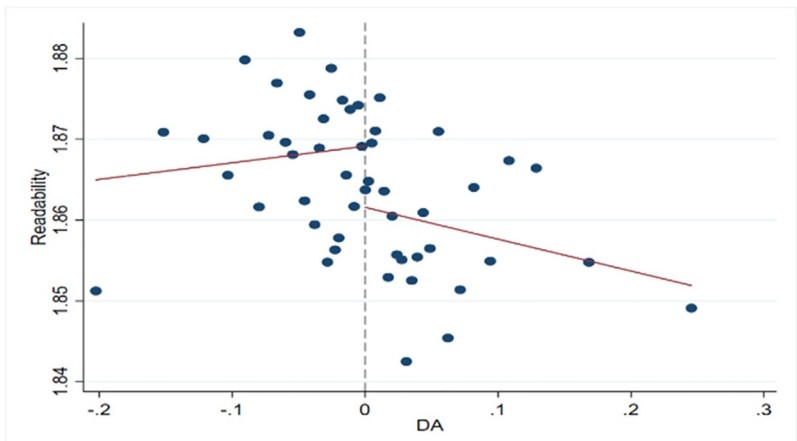

**Fig 1. The relationship between earnings management and the readability of CSR reports as a linear relationship.**

**Table 12. Regression discontinuity of the readability of CSR reports.** The dependent variable is *Readability*. The post indicator variable is DA. All the controlling variables are obtained from the CSMAR database, and t-statistics are reported in parentheses.

| Dep. | Readability |
|---|---|
| **DA** | -2.38e-16*** |
| | (-33.72) |
| **Size** | -0.1020 |
| | (-0.97) |
| **Lev** | 0.0026 |
| | (0.19) |
| **Roe** | -0.0043 |
| | (-0.92) |
| **Cash** | -0.0300 |
| | (-0.47) |
| **SOE** | -0.0256 |
| | (-0.77) |
| **Growth** | -0.0265 |
| | (-0.47) |
| **Dual** | -0.0528** |
| | (-2.36) |
| **Attestation** | -0.0141 |
| | (-1.64) |
| **lwald** | 0.0019 |
| | (0.30) |
| **N** | 5083 |

***, **, and * indicate significance at the 1%, 5%, and 10% levels, respectively

that when managers increase profits through earnings management, they may reduce the readability of Chinese CSR reports by decreasing the readability of Chinese annual reports. When the value of earnings management just exceeds zero, the readability of Chinese CSR reports decreases. When managers reduce the readability of Chinese CSR reports, financing costs are improved and environmental performance is decreased.

## Theoretical and managerial implication

After theoretical analyses and empirical analyses, the following implications are obtained. First, earnings management is negatively related to the readability of Chinese CSR reports. The result indicates that when managers inflate profits through earnings management, they may reduce the readability of Chinese CSR reports. Second, the negative effect is more significant when companies are not punished for violations, when the internal control index of companies is low, when companies lack ISO14001 certification and when companies do not have independent third-party authentication for Chinese CSR reports. Third, further evidence shows that when managers increase profits through earnings management, they may reduce the readability of Chinese CSR reports by decreasing the readability of Chinese annual reports. Fourth, when managers reduce the readability of Chinese CSR reports, debt financing costs and equity financing costs are improved because of information asymmetry. When managers reduce the readability of Chinese CSR reports, CSR performance may be decreased. Fifth, when earnings management just exceeds zero, the readability of Chinese CSR reports decreases. Sixth, the study applies theory of behavior consistency, principal-agent theory and

impression management theory to the readability of Chinese CSR reports. Seventh, the paper expands the study of the readability of Chinese CSR reports.

## Suggestions

To improve the quality of information disclosure of listed companies, enhance information transparency and protect the interests of investors, the recommendations are as follows: First, the government should issue CSR reporting standards to reduce the manipulation of Chinese CSR reports. For example, the government should introduce language disclosure norms for Chinese CSR reports to prevent management from using readability for impression management. Second, Chinese CSR reports disclosed by listed companies must be audited by independent third parties to enhance the credibility of the information. Third, the company needs to strengthen its external and internal supervision to reduce the manipulation space for the readability of Chinese CSR reports.

## Limitations and future research

The limitations of this study are as follows: Firstly, the template of this study is only limited to Chinese reports. Our study will expand the research template to CSR reports from various countries in the future. Secondly, with the development of computer technology, the readability measurement of Chinese CSR reports in the future may become more scientific. Thirdly, psychological research on corporate management may be applied to text analysis research in the further.

## Supporting information

**S1 Appendix. Variable definitions.**
(PDF)

## Acknowledgments

The authors wish to thank the editor and anonymous reviewers who make valuable suggestions for revision and contributions to significantly enhancing the quality of this work.

## Author Contributions

**Writing – original draft:** Bangqi Ren.

**Writing – review & editing:** Bangqi Ren, Sheng Yao.

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
