## [Decision Letter · Decision Letter 0]

2 Nov 2023

PONE-D-23-29697Earnings management and readability of CSR report: Evidence from ChinaPLOS ONE

Dear Dr. Yao,

Thank you for submitting your manuscript to PLOS ONE. After careful consideration, we feel that it has merit but does not fully meet PLOS ONE’s publication criteria as it currently stands. Therefore, we invite you to submit a revised version of the manuscript that addresses the points raised during the review process.

We look forward to receiving your revised manuscript.

Kind regards,

Wajid Khan

Academic Editor

PLOS ONE

3.Thank you for stating the following financial disclosure: 

 [NO - Include this sentence at the end of your statement: The funders had no role in study design, data collection and analysis, decision to publish, or preparation of the manuscript.]. 

6. PLOS requires an ORCID iD for the corresponding author in Editorial Manager on papers submitted after December 6th, 2016. Please ensure that you have an ORCID iD and that it is validated in Editorial Manager. To do this, go to ‘Update my Information’ (in the upper left-hand corner of the main menu), and click on the Fetch/Validate link next to the ORCID field. This will take you to the ORCID site and allow you to create a new iD or authenticate a pre-existing iD in Editorial Manager. Please see the following video for instructions on linking an ORCID iD to your Editorial Manager account: " ext-link-type="uri" xlink:type="simple">https://www.youtube.com/watch?v=_xcclfuvtxQ".

Reviewers' comments:

Reviewer's Responses to Questions

**Comments to the Author**

1. Is the manuscript technically sound, and do the data support the conclusions?

Reviewer #1: Yes

Reviewer #2: Yes

2. Has the statistical analysis been performed appropriately and rigorously? 

Reviewer #1: Yes

Reviewer #2: Yes

3. Have the authors made all data underlying the findings in their manuscript fully available?

Reviewer #1: Yes

Reviewer #2: Yes

4. Is the manuscript presented in an intelligible fashion and written in standard English?

Reviewer #1: Yes

Reviewer #2: Yes

5. Review Comments to the Author

Reviewer #1: Dear Editor,

I am glad to have read this paper. The writing of the article is relatively standardized and the arguments are very detailed. However, there are also some suggestions for the authors to revise and reference.

(1) The innovative points of the article suggest further clarification, as the negative relationship between report readability and earnings management is easily understood and accepted.

(2) For Chinese reports, as mentioned by the authors, how to objectively evaluate their readability and the scientific nature of this evaluation method are very important. How did the authors specifically implement it and what are the innovations?

(3) What are the specific contributions of the authors' research compared to existing reports? Authors need to supplement and clarify.

Reviewer #2: Manuscript ID: PONE-D-23-29697

Earnings management and readability of CSR report: Evidence from China

Comments to the author

I sincerely congratulate the author(s) for conducting research on such an important topic. However, the author(s) may improve some areas for better clarity. Here are some comments on this matter:

1. In order to improve the quality of the paper, please rewrite the abstract section by methodology part (sample, research method, time period of the study, etc) in a better way.

2. Provide a concise and clear statement of the research objective in the introduction to help readers to understand the main focus of the study

3. The authors are required to provide a strong theoretical background below the introduction section of the study separately.

4. The author has included a few recent publications in his/ her literature review while shaping his/ her arguments in the manuscript. Therefore, citing more recent literature might help the author to strengthen the hypothesis development.

5. In the literature review section, the research gap is missing. So, provide a small paragraph just below the literature review.

6. The author needs to provide clarity behind the uses of control variable and its impact on the dependent variable of the study.

7. The findings of the paper have definitely added value to the existing literature in the domain of CSR.

8. The authors are asked to develop the conclusion section by including the unique contributions of the paper, theoretical and managerial implications, limitations of the research, and future research directions.

9. Please cite relevant references wherever necessary.

a) Pareek, R., Sahu, T. N. (2021). How far the ownership structure is relevant for CSR performance? An empirical investigation. Corporate Governance: The International Journal of Business in Society, 22(1), 128-147. https://doi.org/10.1108/CG-10-2020-0461

b) Ghosh, S., Pareek, R., Sahu, T.N. (2022). Does board size matter for CSR performance? A GMM-based dynamic panel data approach. International Journal of Business Excellence, Vol. ahead-of-print No. ahead-of-print. https://doi.org/ 10.1504/ijbex.2022.10049821

c) Agarwala, N., Pareek, R., Sahu, T. N. (2023). Do Firm Attributes Impact CSR Participation? Evidence From a Developing Economy. International Journal of Emerging Markets. Vol. ahead-of-print No. ahead-of-print. https://doi.org/10.1108/IJOEM-05-2022-0876.

d) Ghosh, S., Pareek, R., Sahu, T.N. (2022). The Role of Corporate Governance and Company Specific Characteristics on Environmental Disclosure Practices in India. NIMMS Management Review, 30(4), 64-89. https://doi.org/10.53908/NMMR.300404

e) Pareek, R., Sahu, T.N. (2023) The nonlinear effect of executive compensation on corporate social responsibility performance. Rajagiri Management Journal, Vol. ahead-of-print No. ahead-of-print. https://doi.org/10.1108/RAMJ-06-2022-0094

f) Almahrog, Y., Ali Aribi, Z., Arun, T. (2018). Earnings management and corporate social responsibility: UK evidence. Journal of Financial Reporting and Accounting, 16(2), 311-332.

g) Ehsan, S., Nurunnabi, M., Tahir, S., Hashmi, M. H. (2020). Earnings management: A new paradigm of corporate social responsibility. Business and Society Review, 125(3), 349-369.

6. PLOS authors have the option to publish the peer review history of their article (what does this mean?). If published, this will include your full peer review and any attached files.

Reviewer #1: **Yes: **Zheng-Yong Zhang

Reviewer #2: No

---

## [Author Response · Author response to Decision Letter 0]

13 Jan 2024

Response to Referee 1 On

Thank you for your comments. Responding to them has enabled us to significantly improve the paper. The remainder of this memo explains our response in detail. We first quote each of your comments and then provide our response.

Comment 1: The innovative points of the article suggest further clarification, as the negative relationship between report readability and earnings management is easily understood and accepted.

Response: Thank you very much for your constructive suggestions. In the innovation sections of the introduction, we state that: “Lo, Ramos, and Rogo [1] proved that the relationship between earnings management and the readability of American annual reports was negative. Ye and Wang [35] proved that the relationship between earnings management and the readability of Chinese annual reports was negative. Whether in the United States or China, managers may reduce the readability of annual reports to prevent investors from detecting artificially increased profits through earnings management. However, few studies focus on the negative relationship between earnings management and the readability of Chinese CSR reports. Therefore, we want to study the issue and extend previous research. Using theoretical and empirical methods to prove that when managers increase profits through earnings management, the readability of Chinese CSR reports may be reduced.

Comment 2: For Chinese reports, as mentioned by the authors, how to objectively evaluate their readability and the scientific nature of this evaluation method are very important. How did the authors specifically implement it and what are the innovations?

Response: This is a good comment and we appreciate your advice. The readability of Chinese reports is different from the readability of English reports. Because there is a huge difference between Chinese and English. Therefore, we may not use Fog and Flesh to measure the Chinese readability. However, scientific and reasonable measurement of Chinese readability is crucial for our research. Generally speaking, the readability of the Chinese CSR reports is good when a text contains simpler sentences. Simple sentences usually contain fewer characters in a single sentence and fewer vocabularies in a single sentence. The readability of the Chinese CSR reports is poor when a text contains more complex sentences. Complex sentences usually contain more characters in a single sentence and more vocabularies in a single sentence. When Chinese CSR reports contain more pages, it means that more detailed information is disclosed. The readability of Chinese CSR reports is good when a text contains more pages. After homogenizing and standardizing the average character of a single sentence, the average vocabulary of a single sentence and the total number of pages, the readability of Chinese CSR reports is constructed. To prove the science of the measurement method, it has been applied to many studies in China.

Comment 3: What are the specific contributions of the authors' research compared to existing reports? Authors need to supplement and clarify.

Response: We appreciate your insightful comments very much. Exiting literature proved that the relationship between earnings management and the readability of American annual reports was negative [1]. Ye and Wang [35] proved that the relationship between earnings management and the readability of Chinese annual reports was negative. In the section of the introduction, we state that: “However, few studies focus on the negative relationship between earnings management and the readability of Chinese CSR reports. Therefore, we want to study the issue and extend previous research.”

Thank you again for your help and we welcome additional comments that you may have.

 

Response to Referee 2 On

Thank you for your comments. Responding to them has enabled us to significantly improve the paper. The remainder of this memo explains our response in detail. We first quote each of your comments and then provide our response.

Comment 1: In order to improve the quality of the paper, please rewrite the abstract section by methodology part (sample, research method, time period of the study, etc) in a better way.

Response: We appreciate this comment. The abstract section has been rewritten in a better way. We state that: “The literature has confirmed that when managers increase profits through earnings management, the readability of annual reports may be reduced [1,35]. Whether this conclusion is suitable for Chinese corporate social responsibility (CSR) reports, however, is still unclear. Based on the panel data of 5083 Chinese non-financial listed companies from 2010 to 2019, this paper adopts multiple linear regression to investigate the impact of earnings management on the readability of Chinese CSR reports. The results show that: (1) There is a significant negative correlation between earnings management and the readability of Chinese CSR reports, with the readability of Chinese annual reports as a mediating variable. (2) The negative effect is more significant when companies are not punished for this violation, when the internal control index is low, when companies lack ISO14001 certification and when companies do not have independently third-party authentication for Chinese CSR reports. (3) When earnings management just exceeds zero, the readability of Chinese CSR reports decreases. (4) The economic consequences of reducing the readability of Chinese CSR reports are that financing costs are increased and environmental performance is decreased. This study extends the negative relationship between earnings management and the readability from annual reports to Chinese CSR reports. To prevent investors from detecting earnings management, the readability of Chinese CSR reports may be reduced. At the same time, the study has definitely added value to the existing literature in the domain of CSR.”

Comment 2: Provide a concise and clear statement of the research objective in the introduction to help readers to understand the main focus of the study.

Response: This is a nice suggestion. In the section of the introduction, we further interpret the main focus of the study. We state that: “Many studies indicate that when managers increase profits through earnings management, they may reduce the readability of annual reports [1,35]. Whether this conclusion is suitable for Chinese CSR reports, however, is still unclear. With the development of Chinese economy, CSR reports are receiving increasing attention. More and more investors obtain business information through reading Chinese CSR reports. As a result, we want to study the relationship between earnings management and the readability of Chinese CSR reports. When managers increase profits through earnings management, how does the readability of Chinese CSR reports change?”

Comment 3: The authors are required to provide a strong theoretical background below the introduction section of the study separately.

Response: We are grateful for the suggestion. Theoretical background is added in the introduction. We state that: “Theory of behavior consistency is that individuals exhibit certain similarities and stability in their behaviors and behavioral styles in different scenarios [2,3]. Chinese annual reports and Chinese CSR reports are different ways of disclosing information. When managers increase profits through earnings management, they may adopt consistent measures to manipulate the readability of Chinese annual reports and the readability of Chinese CSR reports. Existing research has proved that when managers increase profits through earnings management, the readability of American annual reports and the readability of Chinese annual reports may be reduced [1,35]. Based on consistency theory, when managers reduce the readability of Chinese annual reports, the readability of Chinese CSR reports may be decreased. Therefore, when managers increase profits through earnings management, the readability of Chinese CSR reports may be reduced. Based on principal-agent theory, managers have an information advantage and shareholders have an information disadvantage. Chinese CSR reports disclose not only the social responsibility of the companies, but also the business information. Therefore, Chinese CSR reports are becoming an important channel for shareholders to obtain information. To make the right investment decision, shareholders read Chinese CSR reports carefully. Meanwhile, Chinese CSR reports are voluntary in China, largely unregulated, and do not have a widely enforced reporting framework. The readability of Chinese CSR reports is easier to manipulate than Chinese annual reports by managers. When managers artificially increase profits through earnings management, they may reduce the readability of Chinese CSR reports. The purpose is to prevent shareholders from detecting earnings management and increase information asymmetry between managers and shareholders. Based on impression management theory, managers may manipulate the readability of Chinese CSR reports to improve investors’ impression of the company. If investors detect that managers increase profits through earnings management, the impression of the company is negatively affected. To prevent investors from detecting the behavior of earnings management through reading Chinese CSR reports, managers may reduce the readability of Chinese CSR reports. It can give investors a good impression of the company.”

Comment 4: The author has included a few recent publications in his/ her literature review while shaping his/ her arguments in the manuscript. Therefore, citing more recent literature might help the author to strengthen the hypothesis development.

Response: We are grateful for the suggestion. We cite more recent literature to strengthen the hypothesis development in the literature review. We state that: “There are many existing studies on corporate social responsibility and CSR performance. Pareek and Sahu [8] found a non-liner inverted U-shaped relationship between foreign ownership and the CSR performance. Ghosh, Pareek, and Sahu [9] found that board sizes may influence CSR performance. Agarwala, Pareek, and Sahu [10] found an inverse U-shape relationship between companies’ sizes and corporate social responsibility. CSR participation was positively related with small-sized firms, but as the firms became larger in size, their relationship with CSR became negative. Pareek and Sahu [11] found that there was an inverted U-shaped relationship between executive compensation and CSR performance. Ghosh, Pareek, and Sahu [12] found that governance factors like board sizes and board meetings were showing a positive effect on disclosure practices.” in the section of the research on readability. We state that: “Almahrog, Ali Aribi, and Arun [28] proved that there was a negative relationship between the level of CSR and earnings management. Ehsan et al. [29] adopted a systematic approach to review the existing studies on the relationship between corporate social responsibility and earnings management.” in the section of the research on earnings management.

Comment 5: In the literature review section, the research gap is missing. So, provide a small paragraph just below the literature review.

Response: Thank you for your suggestion. We add a small paragraph about the research gap in the literature review. We state that: “With the development of Chinese economy, CSR reports are increasingly valued. Chinese CSR reports not only disclose environmental information but also business information of the companies. Many investors may understand the business situation of the companies through reading Chinese CSR reports. To prevent investors from detecting earnings management, is the readability of Chinese CSR reports reduced? This issue has not been studied yet. Therefore, we are prepared to explore the issue.”

Comment 6: The author needs to provide clarity behind the uses of control variable and its impact on the dependent variable of the study.

Response: Thank you for your suggestion. To explain the reasons for selecting control variables, we state that: “All control variables come from relevant literature on the readability of CSR reports.” in the section of the control variables. To explain its impact on the dependent variables, we state that: “In addition, we treat year and firm as dummy variables in the regressions to control for year and firm fixed effects, respectively. Besides, we present the expected sign between each variable and the readability of Chinese CSR reports based on the related literature, where “+” represents a positive correlation, “-” represents a negative correlation, and “+/-” represents an uncertain sign. See appendix for variable definitions.” in the section of the control variables.

Comment 7: The findings of the paper have definitely added value to the existing literature in the domain of CSR.

Response: We are grateful for the suggestion. The study of the readability of CSR reports is part of the study of corporate social responsibility. When elaborating on research contributions, we state that: “The study has definitely added value to the existing literature in the domain of CSR.” in the section of the abstract.

Comment 8: The authors are asked to develop the conclusion section by including the unique contributions of the paper, theoretical and managerial implications, limitations of the research, and future research directions.

Response: We are grateful for the suggestion. To clarify the unique contributions of the paper, we state that: “The study of the readability of Chinese CSR reports has definitely added value to the existing literature in the domain of CSR. Existing literature proved that managers decreased the readability of annual reports to prevent investors from detecting artificially increased profits through earnings management [1,35]. We extend the conclusion of the study from annual reports to Chinese CSR reports.” To interpret theoretical and managerial implications, we state that: “Sixth, the study applies theory of behavior consistency, principal-agent theory and impression management theory to the readability of Chinese CSR reports. Seventh, the paper expands the study of the readability of Chinese CSR reports.” To illustrate the limitations of the research and future research directions, we state that: “The limitations of this study are as follows: Firstly, the template of this study is only limited to Chinese reports. Our study will expand the research template to CSR reports from various countries in the future. Secondly, with the development of computer technology, the readability measurement of Chinese CSR reports in the future may become more scientific. Thirdly, psychological research on corporate management may be applied to text analysis research in the further.”

Comment 9: Please cite relevant references wherever necessary.

Response: Thank you for your suggestion. We have cited all the provided references in the literature review. 

Thank you again for your help and we welcome additional comments that you may have.

---

## [Decision Letter · Decision Letter 1]

8 Feb 2024

PONE-D-23-29697R1Earnings management and readability of CSR report: Evidence from ChinaPLOS ONE

Dear Dr. Yao,

Thank you for submitting your manuscript to PLOS ONE. After careful consideration, we feel that it has merit but does not fully meet PLOS ONE’s publication criteria as it currently stands. Therefore, we invite you to submit a revised version of the manuscript that addresses the points raised during the review process.

We look forward to receiving your revised manuscript.

Kind regards,

Wajid Khan

Academic Editor

PLOS ONE

Journal Requirements:

Reviewers' comments:

Reviewer's Responses to Questions

**Comments to the Author**

1. If the authors have adequately addressed your comments raised in a previous round of review and you feel that this manuscript is now acceptable for publication, you may indicate that here to bypass the “Comments to the Author” section, enter your conflict of interest statement in the “Confidential to Editor” section, and submit your "Accept" recommendation.

Reviewer #1: All comments have been addressed

Reviewer #2: All comments have been addressed

2. Is the manuscript technically sound, and do the data support the conclusions?

Reviewer #1: Yes

Reviewer #2: Yes

3. Has the statistical analysis been performed appropriately and rigorously? 

Reviewer #1: Yes

Reviewer #2: Yes

4. Have the authors made all data underlying the findings in their manuscript fully available?

Reviewer #1: Yes

Reviewer #2: No

5. Is the manuscript presented in an intelligible fashion and written in standard English?

Reviewer #1: Yes

Reviewer #2: Yes

6. Review Comments to the Author

Reviewer #1: The research article has been revised and I personally believe that it meets the requirements for publication and can be considered for publication.

Reviewer #2: Manuscript ID: PONE-D-23-29697R1

Earnings management and readability of CSR report: Evidence from China

Comments to the author

I sincerely congratulate the author(s) for conducting research on such an important topic. However, the author(s) may improve some areas for better clarity. Here are some comments on this matter:

1. In the abstract section the policy recommendation part is missing, if possible, try to incorporate it.

2. Authors must segregate the introduction and theoretical background separately to make the readers to understand this article easily without any confusion.

7. PLOS authors have the option to publish the peer review history of their article (what does this mean?). If published, this will include your full peer review and any attached files.

Reviewer #1: **Yes: **Zheng-Yong Zhang

Reviewer #2: No

---

## [Author Response · Author response to Decision Letter 1]

14 Feb 2024

Response1: Thank you very much for your constructive suggestions. In the abstract section, I add that: “To improve the quality of information disclosure of listed companies, the recommendations are as follows: First, the government should issue CSR reporting standards to reduce the manipulation of Chinese CSR reports. Second, Chinese CSR reports disclosed by listed companies must be audited by independent third parties to enhance the credibility of the information. Third, the company needs to strengthen its external and internal supervision to reduce the manipulation space for the readability of Chinese CSR reports.”

Response2: This is a good comment and we appreciate your advice. In the introduction, I delete that: “theory of behavior consistency is that individuals exhibit certain similarities and stability in their behaviors and behavioral styles in different scenarios. Chinese annual reports and Chinese CSR reports are different ways of disclosing information. When managers increase profits through earnings management, they may adopt consistent measures to manipulate the readability of Chinese annual reports and the readability of Chinese CSR reports. Existing research has proved that when managers increase profits through earnings management, the readability of American annual reports and the readability of Chinese annual reports may be reduced . Based on consistency theory, when managers reduce the readability of Chinese annual reports, the readability of Chinese CSR reports may be decreased. Therefore, when managers increase profits through earnings management, the readability of Chinese CSR reports may be reduced. Based on principal-agent theory, managers have an information advantage and shareholders have an information disadvantage. Chinese CSR reports disclose not only the social responsibility of the companies, but also the business information. Therefore, Chinese CSR reports are becoming an important channel for shareholders to obtain information. To make the right investment decision, shareholders read Chinese CSR reports carefully. Meanwhile, Chinese CSR reports are voluntary in China, largely unregulated, and do not have a widely enforced reporting framework. The readability of Chinese CSR reports is easier to manipulate than Chinese annual reports by managers. When managers artificially increase profits through earnings management, they may reduce the readability of Chinese CSR reports. The purpose is to prevent shareholders from detecting earnings management and increase information asymmetry between managers and shareholders. Based on impression management theory, managers may manipulate the readability of Chinese CSR reports to improve investors’ impression of the company. If investors detect that managers increase profits through earnings management, the impression of the company is negatively affected. To prevent investors from detecting the behavior of earnings management through reading Chinese CSR reports, managers may reduce the readability of Chinese CSR reports. It can give investors a good impression of the company.”

Response3: I have made all data underlying in my manuscript fully available. ICPS database requires registering an account to download data.

---

## [Decision Letter · Decision Letter 2]

22 Feb 2024

PONE-D-23-29697R2Earnings management and readability of CSR report: Evidence from ChinaPLOS ONE

Dear Dr. Yao,

Thank you for submitting your manuscript to PLOS ONE. After careful consideration, we feel that it has merit but does not fully meet PLOS ONE’s publication criteria as it currently stands. Therefore, we invite you to submit a revised version of the manuscript that addresses the points raised during the review process.

We look forward to receiving your revised manuscript.

Kind regards,

Wajid Khan

Academic Editor

PLOS ONE

Journal Requirements:

Reviewers' comments:

Reviewer's Responses to Questions

**Comments to the Author**

1. If the authors have adequately addressed your comments raised in a previous round of review and you feel that this manuscript is now acceptable for publication, you may indicate that here to bypass the “Comments to the Author” section, enter your conflict of interest statement in the “Confidential to Editor” section, and submit your "Accept" recommendation.

Reviewer #2: All comments have been addressed

2. Is the manuscript technically sound, and do the data support the conclusions?

Reviewer #2: Yes

3. Has the statistical analysis been performed appropriately and rigorously? 

Reviewer #2: Yes

4. Have the authors made all data underlying the findings in their manuscript fully available?

Reviewer #2: Yes

5. Is the manuscript presented in an intelligible fashion and written in standard English?

Reviewer #2: Yes

6. Review Comments to the Author

Reviewer #2: Manuscript ID: PONE-D-23-29697R2

Earnings management and readability of CSR report: Evidence from China

Comments to the author

I sincerely congratulate the author(s) for conducting research on such an important topic. However, the author(s) may improve some areas for better clarity. Here are some comments on this matter:

1. In conclusion part try to summarise your whole study in few sentences like introduction, methodology, findings.

2. Put the limitations and future research work at the end after suggestions part.

7. PLOS authors have the option to publish the peer review history of their article (what does this mean?). If published, this will include your full peer review and any attached files.

Reviewer #2: No

---

## [Author Response · Author response to Decision Letter 2]

23 Feb 2024

Response1: Thank you very much for your constructive suggestions. In conclusion part, we state that: “When managers increase profits through earnings management, they may reduce the readability of Chinese CSR reports. The negative effect is more significant when companies are not punished for violations, when the internal control index of companies is low, when companies lack ISO14001 certification and when companies do not have independent third-party authentication for Chinese CSR reports. Further study shows that when managers increase profits through earnings management, they may reduce the readability of Chinese CSR reports by decreasing the readability of Chinese annual reports. When the value of earnings management just exceeds zero, the readability of Chinese CSR reports decreases. When managers reduce the readability of Chinese CSR reports, financing costs are improved and environmental performance is decreased.”

Response2: This is a good comment and we appreciate your advice. I put the limitations and future research work at the end after suggestions part.

---

## [Decision Letter · Decision Letter 3]

12 Mar 2024

Earnings management and readability of CSR report: Evidence from China

PONE-D-23-29697R3

Dear Dr. Yao,

We’re pleased to inform you that your manuscript has been judged scientifically suitable for publication and will be formally accepted for publication once it meets all outstanding technical requirements.

Kind regards,

Wajid Khan

Academic Editor

PLOS ONE

Additional Editor Comments (optional):

Reviewers' comments:

Reviewer's Responses to Questions

**Comments to the Author**

1. If the authors have adequately addressed your comments raised in a previous round of review and you feel that this manuscript is now acceptable for publication, you may indicate that here to bypass the “Comments to the Author” section, enter your conflict of interest statement in the “Confidential to Editor” section, and submit your "Accept" recommendation.

Reviewer #2: All comments have been addressed

2. Is the manuscript technically sound, and do the data support the conclusions?

Reviewer #2: Yes

3. Has the statistical analysis been performed appropriately and rigorously? 

Reviewer #2: Yes

4. Have the authors made all data underlying the findings in their manuscript fully available?

Reviewer #2: Yes

5. Is the manuscript presented in an intelligible fashion and written in standard English?

Reviewer #2: No

6. Review Comments to the Author

Reviewer #2: (No Response)

7. PLOS authors have the option to publish the peer review history of their article (what does this mean?). If published, this will include your full peer review and any attached files.

Reviewer #2: No

---

## [Editor Report · Acceptance letter]

22 Mar 2024

PONE-D-23-29697R3 

PLOS ONE

Dear Dr. Yao, 

I'm pleased to inform you that your manuscript has been deemed suitable for publication in PLOS ONE. Congratulations! Your manuscript is now being handed over to our production team.

Kind regards, 

on behalf of

Dr. Wajid Khan 

Academic Editor

PLOS ONE